# Ammonia Bioremediation from Aquaculture Wastewater Effluents Using *Arthrospira platensis* NIOF17/003: Impact of Biodiesel Residue and Potential of Ammonia-Loaded Biomass as Rotifer Feed

**DOI:** 10.3390/ma14185460

**Published:** 2021-09-21

**Authors:** Mohamed Ashour, Ahmed E. Alprol, Ahmed M. M. Heneash, Hosam Saleh, Khamael M. Abualnaja, Dalal Alhashmialameer, Abdallah Tageldein Mansour

**Affiliations:** 1National Institute of Oceanography and Fisheries (NIOF), Cairo 11516, Egypt; ah831992@gmail.com (A.E.A.); aheneash@yahoo.com (A.M.M.H.); husam_aldien2@hotmail.com (H.S.); 2Department of Chemistry, College of Science, Taif University, P.O. Box 11099, Taif 21944, Saudi Arabia; k.ala@tu.edu.sa (K.M.A.); Dsamer@tu.edu.sa (D.A.); 3Animal and Fish Production Department, College of Agricultural and Food Sciences, King Faisal University, P.O. Box 420, Al-Ahsa 31982, Saudi Arabia; 4Fish and Animal Production Department, Faculty of Agriculture (Saba Basha), Alexandria University, Alexandria 21531, Egypt

**Keywords:** *Arthrospira platensis* NIOF17/003, lipid-free biomass, aquaculture wastewater treatment, ammonia adsorption, IR, FTIR, isotherms study, kinetic study, rotifer

## Abstract

The present work evaluated the capability of *Arthrospira platensis* complete biomass (ACDW) and the lipid-free biomass (LFB) to remove ammonium ions (NH_4_^+^) from aquaculture wastewater discharge. Under controlled conditions in flasks filled with 100 mL of distilled water (synthetic aqueous solution), a batch process ion-exchange was conducted by changing the main parameters including contact times (15, 30, 45, 60, 120, and 180 min), initial ammonium ion concentrations (10, 20, 30, 40, 50, and 100 mg·L^−1^), and initial pH levels (2, 4, 6, 8, and 10) at various dosages of ACDW and LFB as adsorbents (0.02, 0.04, 0.06, 0.08, and 0.1 g). After lab optimization, ammonia removal from real aquaculture wastewater was also examined. The removal of ammonium using ACDW and LFB in the synthetic aqueous solution (64.24% and 89.68%, respectively) was higher than that of the real aquaculture effluents (25.70% and 37.80%, respectively). The data of IR and Raman spectroscopy confirmed the existence of various functional groups in the biomass of ACDW and LFB. The adsorption equilibrium isotherms were estimated using Freundlich, Langmuir, and Halsey models, providing an initial description of the ammonia elimination capacity of *A. platensis*. The experimental kinetic study was suitably fit by a pseudo-second-order equation. On the other hand, as a result of the treatment of real aquaculture wastewater (RAW) using LFB and ACDW, the bacterial counts of the LFB, ACDW, ACDW-RAW, and RAW groups were high (higher than 300 CFU), while the LFB-RAW group showed lower than 100 CFU. The current study is the first work reporting the potential of ammonia-loaded microalgae biomass as a feed source for the rotifer (*Brachionus plicatilis*). In general, our findings concluded that *B. plicatilis* was sensitive to *A. platensis* biomass loaded with ammonia concentrations. Overall, the results in this work showed that the biomass of *A. platensis* is a promising candidate for removing ammonia from aquaculture wastewater.

## 1. Introduction

The increase in population in recent years has led to the continuous development and expansion of aquaculture activities, which have directly led to an increase in water consumption, causing many problems, especially the issue of wastewater disposal. This wastewater usually contains a high content of nitrogen, which comes mainly from fish excreta and feed residue. To date, several biological and chemical methods have been used in the treatment of aquaculture wastewater. However, these methods have several disadvantages such as being no environmentally friendly, inefficient, and without economic value on large-scale application [1,2,3,4]. In general, this aquaculture waste is laden with various types of pollutants, especially high quantities of nitrogenous compounds, mainly in ammonia form (NH_4_^+^). These nitrogenous compounds, together with other pollutants, cause a high level of eutrophication along the coastal areas, which results in the growth of invasive organisms [5]. Recently, there has been global interest in the removal of pollutants from different types of wastewater attributed to the high potency of these pollutants to contaminate food and water sources in addition to establishing appropriate growth parameters for several pathogenic microorganisms [6].

Nitrogen is a main chemical compound for life and industrial use, while ammonia is an integral component of the nitrogen cycle life and is a source of free nitrogen. Industrial wastewater, agricultural activities, and municipal effluents increase the ammonia nitrogen discharges into environmental resources [7]. Ammonium pollution affects the aquatic quality of water bodies, resulting in severe environmental problems such as pH shift, cyanotoxin creation, oxygen reduction, eutrophication of downstream liquids, enhanced eutrophication of rivers and lakes, and exhaustion of dissolved oxygen, being toxic aquatic animals at a level greater than 1.9 mg·L^−1^ [8,9]. 

In aquaculture wastewaters, ammonia concentration should be decreased to tolerable limits before being discharged into the aquatic environment [10]. Thus, wastewater treatment is aimed at the removal of pollutants and includes may methods such as biological processes (anaerobic and aerobic), chemical oxidation, nanofiltration, combustion, ultrafiltration, precipitation, flocculation, reverse osmosis, evaporation, and adsorption [11]. Biological treatments are used for the traditional method of removing ammonia from industrial and municipal wastewaters. However, toxic shock, pH changes, low dissolved oxygen, and cold temperatures in the winter can all impede this approach [12]. One of the most effective ways to decrease the amount of aquaculture wastewater contaminants is to develop an adsorbent that is a natural component [13].

Recently, microalgae have been discussed as a valuable feedstock for many applications, including biofuel production [14,15]. It was calculated that roughly 40–100 kg of inorganic nitrogen compounds and 11–13 ML/Ha/year of water are required for 1 ton of microalgal growth [8]. It was also anticipated that roughly 2500 m^3^ of wastewater might be processed to produce 1 ton of microalgal biomass [16]. As a result, using wastewater for algal cultivation could allow for the removal of nutrients from effluents while also reducing water consumption by 90%. Furthermore, by employing mixotrophic cultivation, the negative impact of high ammonium concentrations on microalgal growth and biomass yield will be reduced. In fact, mixotrophic growth offers more energy for rapid ammonium assimilation, lowering the ammonium inhibition and increasing the microalgal biomass yield [8]. 

Algal biomass is a natural source of bioactive substances that can be employed in a variety of applications such as aquaculture, biofertilizer, food supplement, cosmetics, biodiesel, antimicrobial activities, and bioremediation. *Arthrospira platensis*, a cyanophyte, has emerged as one of the most promising agents for the production of possibly novel chemicals. It has been shown to create intracellular and extracellular metabolites with a variety of biological activities, including antifungal, antiviral, and antibacterial activities [17,18]. Moreover, *A. platensis* displays a broad spectrum of antimicrobial and antioxidant properties against Gram-positive and Gram-negative pathogens [19]. Functional groups such as hydroxyl, carboxyl, phosphate, and sulfate, as well as other charged groups, can interfere with pollutant binding sites in the biomass. This microalgal species has been used to remove dyes and heavy metals from aqueous solutions with great success [20]. 

In the last few years, microalgal biomass has been studied as an important adsorbent for the purification of aquaculture wastewater, owing to its capability of applying and/or accumulating heavy metals, nutrients, and various materials in their cells. Adsorption is a reducing technology that removes hazardous pollutants from aqueous solutions using living or dead biomasses. Adsorption is a multistep process that includes ion exchange, chelation, physical adsorption, and trapping in inter- and intra-fibrillar capillaries and the space of the structural polysaccharide network as a result of concentration gradients and diffusion [21]. The use of microalgal biomass for wastewater treatment requires the least amount of mechanical equipment and consumes less energy compared to other conventional methods [7]. Many studies have confirmed that microalgal biomass has great potential for N elimination and stated effective cultivations [22,23,24,25,26,27,28,29]. In the study by Zaki et al. [30], a cyanobacterium strain *A. platensis* NIOF17/003 (GenBank number: MW396472) was isolated, molecularly identified, and sub-cultured under controlled conditions. The growth rate curve, biochemical composition, lipid profile, and biodiesel production potential of this strain were investigated. Recently, *A. platensis* NIOF17/003 showed a positive effect on the bioremediation of organic dye (Ismate violet 2R, IV2R) from industrial textile effluents [31]. However, the ability of this strain to adsorb nitrogen from wastewater and the impact of nitrogen-rich biomass as rotifer feed need further investigation. The present study aimed to investigate the potential application of *A. platensis* NIOF17/003, either as complete dry biomass (ACDB) or as lipid-free biomass (LFB), for the removal of ammonium ions (NH_4_^+^) from aquaculture wastewater (under laboratory-controlled conditions and using real aquaculture effluent). The effect of ACDB and LFB on the bacterial communities of the aquaculture effluent was evaluated. Furthermore, the application of ACDB and LFB, after saturation with nitrogen, as a feed for a marine rotifer, *B. plicatilis,* was examined.

## 2. Materials and Methods

### 2.1. Arthrospira Strain

In the current study, *Arthrospira platensis* NIOF17/003 (GenBank number: MW396472) was isolated (from a saline-alkaline lake, El-Khadra Lake, Wadi El-Natrun, Egypt), genetically identified, cultured under controlled conditions, and appreciated as a potential source for biodiesel production, as previously described by Zaki et al. [30]. Briefly, batch culture of *A. platensis* NIOF17/003 was cultivated for 2 weeks in 0.5 L of sterile Zarrouk culture medium [32], under constant controlled cultivation conditions of temperature (28.5 ± 1.5 °C), illumination (4000 ± 500 Lux·24 h^−1^), and aseptic aeration, with shaking at 80 rpm [30]. Biomass of *A. platensis* was harvested by centrifugation (3000× *g*·10 min^−1^) on day 8, which was the late exponential phase (LEP), to evaluate the biochemical composition, lipid profile, and physicochemical characteristics of the fatty acid methyl esters (FAMEs) identified and calculated according to the procedures reported by Zaki et al. [30]. Furthermore, the ACDW and LFB of *A. platensis* were individually dried for 48 h at 55 °C and preserved in vacuum bags for further experiments. 

### 2.2. Adsorption Experiments 

#### 2.2.1. Adsorbent Preparation

In the current study, ACDW and LFB were used as adsorbents for ammonium ions (NH_4_^+^) from aquaculture wastewater. Batch process ion exchange was achieved using flasks (100 mL of distilled water) in a shaker under different parameters of contact times (15, 30, 45, 60, 120, and 180 min), initial ammonium ion concentrations (10, 20, 30, 40, 50, and 100 mg·L^−1^), and initial ammonium ion pH levels (2, 4, 6, 8, and 10) at various dosages (0.02, 0.04, 0.06, 0.08, and 0.1 g) of ACDW or LFB. The solution was shaken at 120 rpm at room temperature (25 °C) at constant volume of solution. Ammonium ion concentrations were prepared using stock solution of ammonium chloride (NH_4_Cl) solution. The suspended solids and adsorbents were filtered, and the filtrate was analyzed according to APHA [33]. Water samples for ammonia (NH_4_-N) determination were placing in dark brown bottles as described by Koroleff in Grasshoff [34] and the reagents (1 mL of citrate solution (480 g·L^−1^), 1 mL of phenol reagent (38 g·L^−1^), and 1 mL of hypochlorite reagent (5%)) were added immediately to 35 mL of the sample in the field. The mixture was allowed to stand overnight (10–12 h), the blue color of indophenols formed was measured using a spectrophotometer at 630 nm, and the results were expressed as g·L^−1^.

The removal percentage of NH_4_^+^ and the amount of NH_4_^+^ retained by ACDW and LFB were using by Equations (1) and (2), respectively [35].
(1)$$\mathrm{Removal}\;\mathrm{percentage}\;\left(\%\right)=\frac{\left(C_{i}-C_{f}\right)}{C_{i}}\times 100,$$
(2)$$q_{e}=\frac{\left(C_{i}-C_{f}\right)\times V}{M},$$
where C_i_ (mg·L^−1^) represents the initial concentration, C_f_ (mg·L^−1^) represents the equilibrium concentration, V (L) represents the total volume of ammonium, and M (grams) represents the weight of ACDW or LFB.

#### 2.2.2. Isotherm Studies

Under laboratory conditions (temperature, 25 °C; contact time, 120 min; pH, 6; 0.04 g of adsorbent (ACDW or LFB)), the adsorption isotherm study was applied. The Freundlich, Langmuir, and Halsey models were applied. The Freundlich model describes how the adsorbed ions are retained in many layers and is used to demonstrate heterogeneous systems and reversible sorption processes [36]. The Langmuir model predicts the presence of monolayer coverage of the sorbate at its external surface based on the assumption that intermolecular powers decrease rapidly with distance and predicts monolayer sorption on the external surface of the adsorbent compound [37]. However, the Halsey isotherm model is appropriate for multilayer adsorption and can be fitted to heteroporous substances [38]. Equations (3)–(5) and (7) are the mathematical equations of the Freundlich, Langmuir, and Halsey isotherm models, respectively.
(3)$$q_{e}=K_{f}C_{e}^{1/n},$$
(4)$$\log q_{e\;}=\log K_{f\;}+\frac{1}{n}{\;\mathrm{logC}}_{e},$$
where n is the adsorption intensity, and K_f_ is the Freundlich constant.
(5)$$q_{e}=\frac{Q_{m}K_{a}C_{e}}{1+K_{a}C_{e}},$$
(6)$$\frac{1}{Q_{e}}=\frac{1}{{\mathrm{bq}}_{\max}}\times \frac{1}{C_{e}}+\frac{1}{q_{\max}},$$
where $q_{e}$ is the adsorption capacity at equilibrium, $q_{\max}$ is the maximum sorption capacity, and b is the Langmuir constant.
(7)$${\mathrm{Ln}\;q}_{e}=\frac{1}{n}\mathrm{Ln}\;K+\frac{1}{n}\;\mathrm{Ln}\;\mathrm{Ce},$$
where n and K are Halsey constants.

#### 2.2.3. Kinetic Study

Experiments were conducted in conical flasks at pH 6 by mixing 0.1 g of adsorbents (ACDW or LFB) with 50 mL of an ammonia mixture of 10 mg·L^−1^ concentration. The solution was stirred at room temperature for the required time intervals of 15, 30, 45, 60, 120, and 180 min.

##### Pseudo-First-Order Kinetic Model

The following equation is the linear form of the generalized pseudo-first-order equation [39]:Dq/d_t_ = K_1_ (q_e_ − q_t_),(8)
where q_e_ is the amount of ammonia adsorbed at equilibrium (mg·g^−1^), q_t_ denotes the amount of ammonia adsorbed at time t (mg·g^−1^), and K_1_ denotes the pseudo-first-order rate constant (min^−1^). 

The following criteria were used to evaluate the integrating equation:Log (q_e_/q_e_ − q_t_) = k_1_t/2.303,(9)

In a linear equation, the pseudo-first-order equation is provided by the following formula: Log (q_e_ − q_t_) = log q_e_ − k_1_t/2.303.(10)

Plots of log (q_e_ − q_t_) against (t) should yield a linear relationship between k1 and qe, which can be assessed using the slope and intercept, respectively.

##### Pseudo-Second-Order Kinetic Model 

The pseudo-second order equation was expressed as the following form [40]:dq_t_/d_t_ = K_2_(q_e_ − q_t_)^2^,(11)
where K_2_ indicates the constant of second-order rate (g·mg^−1^·min). 

The integrating equation was as follows:1/(q_e_ − q_t_) = 1/q_e_ + K_2_.(12)

Ho et al. [41] obtained a linear form of the pseudo-second-order equation as follows:t/q_t_ = 1/K_2_q_e_^2^ + t/q_e_.(13)

Plots of (t/q_t_) against (t) should yield a linear relationship, with the values of q_e_ and K_2_ parameters calculated using the slope and intercept, respectively.

#### 2.2.4. Application of Aquaculture Wastewater

A real aquaculture effluent sample was obtained from a private fish farm located in Alexandria (31° 12′ 30.07″ N and 29° 58′ 41.66″ E), Egypt. The aquacultured fish species was Nile tilapia, *Oreochromis niloticus.* The physicochemical parameters of effluent sample of pH (8.7), total suspended solids (TSS, 211 mg·L^−1^), total dissolved solids (TDS, 3110 mg·L^−1^), and ammonium ions (NH_4_^+^, 13.93 μg·L^−1^) were determined. ACDW and LFB were used to remove ammonia from aquaculture wastewater effluent under the ideal conditions of pH 8 for 120 min (contact time) and 0.1 g of ACDW and LFB (adsorbents). As the ammonia concentration was low in the aquaculture effluent sample, a volume of ammonia solution was added to a final concentration of 10 mg·L^−1^, depending on the results obtained from laboratory conditions. To remove precipitates and suspended matter, the solution was filtered. To quantify the effect of adsorbent on ammonia removal, deionized water with a similar concentration of ammonia was prepared as a control.

#### 2.2.5. Adsorbent Characterization

To evaluate the morphological features of ACDW and LFB before and after ammonia removal, scanning electron microscopy (SEM), Fourier-transform infrared (FTIR) spectroscopy, and Raman analytical spectrophotometry were employed using a JSM-IT200 SEM microscope, a Bruker Model Vertex 70 FTIR spectrometer connected to a platinum ATR unit, Bruker, Germany, and a Bruker Senterra Raman spectrometer (USA), respectively. 

### 2.3. Influence of Adsorbents on the Bacterial Count

The effects of adsorbents (ACDW and LFB) on the structure of the bacterial communities in real aquaculture wastewater effluent after ammonia removal were examined. According to Yahaya et al. [42], the bacterial count was examined in five treatments (samples): (1) filtered lipid-free biomass (LFB), (2) aquaculture wastewater treated with LFB (LFB-RAW), (3) filtered complete dry biomass (ACDW), (4) aquaculture wastewater treated with ACDW (ACDW-RAW), and (5) an aquaculture wastewater (RAW) sample that was not treated with *A. platensis* as a control. Sterile nutrient agar media was prepared by autoclaving at 120 °C for 20 min; 1 g·mL^−1^ of each sample was inoculated using pouring technique. Diffusion was allowed to occur in cultured plates for 2 h at room temperature. The plates were incubated in an upright position at 37 °C for 48 h. For each treatment, the bacterial colonies were counted (CFU) and estimated using the following formula:CFU = number of colonies × dilution factor/sample volume (mL).(14)

### 2.4. Bioassay Test

In the current study, the influence of different concentrations (0.02, 0.05, 0.10, and 0.20 g) of ACDB and LFB loaded with ammonia, compared to the same concentrations without ammonia loading (as a control), on rotifer (*Brachionus plicatilis*, L-type, 180 µm) population, mortality, and females carrying eggs (population and mortality) were investigated. Before the experiment, rotifers were mass produced in culture tanks under controlled culture conditions of temperature (23 °C), salinity (30 ppt), and pH (7.5), using continuous aeration and supplemented with the previously identified marine microalga *Nannochloropsis oceanica* NIOF15/001 [43] at concentration level of 5.5 × 10^6^ cells·mL^−1^·day^−1^. To begin the experiment, *B. plicatilis* was taken from the culture tanks and starved for 24 h to allow total gut discharge before being placed into plastic jars supplied with 500 mL of filtered saltwater, with three replicates per level. This trial was performed for 72 h in the following conditions: temperature (23 °C), salinity (30 ppt), and pH (7.5), without aeration [31]. Tested rotifer parameters of population growth, population mortality, and population and mortality of females carrying eggs were investigated as described previously by Alprol et al. [31]. The population growth was calculated as the increase/decrease in *B. plicatilis* individual number, which was at an initial stocking density of 16,500 ± 540 individuals·L^−1^. The egg-carrying population of *B. plicatilis* was estimated as the number of females carrying eggs, which was at an initial stocking density of 14,000 ± 500 females·L^−1^. On the other hand, the total mortality of *B. plicatilis* (individuals·L^−1^) and the total mortality of females carrying eggs (females·L^−1^) were calculated as the dead organisms and females, respectively, by investigations of samples under optical microscope using a Sedgwick-Rafter counting cell as described previously [30,31]. 

### 2.5. Chemicals

The following reagents were used: ethylenediaminetetraacetic acid (EDTA) (C_10_H_16_N_2_O_8_), zinc sulfate (ZnSO_4_·7H_2_O), citrate solution, phenol reagent, hypochlorite reagent, and ammonium chloride (NH_4_Cl). Solution pH was adjusted with dilute NH_3_ and HNO_3_.

### 2.6. Statistical Analysis

Before performing the statistical analysis, the uncertainties of symmetric, normal, and endogenous data (mean ± SD, *n* = 3) were ascertained. Statistical analyses were applied using SPSS program (IBM, v. 20, Armonk, NY, USA). All evaluated variables were performed at a significant level (*p* < 0.05), to a study ANOVA, followed by Duncan’s multiple range examinations and then least significant difference (LSD) tests.

## 3. Results and Discussion

### 3.1. Characterization of Adsorbents 

#### 3.1.1. FTIR Analysis

Figure 1 illustrates the IR spectra of ACDW and LFB before and after adsorption of ammonia, showing broad and strong absorption peaks at 3268–3877 cm^−1^ (intense and broad band) attributed to the stretching vibration of N–H and O–H groups [44]. The bands observed after adsorption at 2917–2926 cm^−1^ for ACDW and LFB indicate the stretching vibrations of asymmetric C–H bonds of methyl, methylene, and methoxy groups [45]. Furthermore, the presence of new peaks around 2352.58 and 2322.90 cm^−1^ for ACDW and LFB might have been due to the presence of amide. Moreover, the absorption new peaks at 2195.09 cm^−1^ for ACDW indicated the presence of C≡N. The new peak after adsorption for ACWD at 2068.30 cm^−1^ was tentatively assigned to (C≡C) stretching. The absorbance band at 1635.36 and 1632.28 cm^−1^ (before and after adsorption for ACWD), as well as the peaks at 1628.75 and 1626.77 cm^−1^ (before and after adsorption for LFB), could be ascribed to the existence of carbonyl C=O stretching of the carboxyl groups and aromatic C=C ring stretching. Furthermore, NH_2_ group bending was indicated at 1632 and 1626 cm^−^^1^ for ACDW and LFB, respectively [46]. The intense band seen at 1527–1532 cm^−1^ indicated the presence of N–H bending and C=C stretching, which might be attributed to the presence of aromatics. Moreover, there were numerous shoulders and small bands in the scope of the region between 1449.22 and 1233.89 cm^−1^ which were assigned to the aromatic rings, C–O stretching absorption peaks, and C–H bending vibration [47]. Furthermore the signal around 1391–1397 cm^−1^ may have been due to sulfate [45]. The observed peaks at 1032.15 and 1029.56 cm^−1^ before adsorption shifted to around 1439 and 1042.99 cm^−1^ after adsorption of ACDW and LFB, respectively, confirming the presence of C–O stretching. Additionally, the peaks around 448.73 and 457.77 cm^−1^ (before adsorption of ACDW and LFB, respectively) were created by the stretching of C–H. Based on FTIR examination, the creation of new peaks, the alteration in absorption intensity, the disappearance of some peaks, and the shift in wavenumber of functional groups might be attributed to the relationships of different ions with active sites of the biomass. Furthermore, interactions of ammonium ions were expected to occur with NH_2_, C=O, COO, and OH groups on the aromatic groups in the adsorbents [46]. 

#### 3.1.2. Raman Spectral Analysis

Figure 2 shows the vibrational data of the symmetry of chemical bonds and molecules which could be recognized from Raman spectroscopy analysis compositions of *A. platensis* (ACDW and LFB before and after adsorption). Each type of biomolecule was shown to have its own characteristic signature Raman spectrum. The band at 3361 cm^−1^ was ascribed to the N–H stretching, while the absorption peaks at 3640–4181 cm^−1^ were attributed to υ (O–H). The peaks at 1820–1893 cm^−1^ were related to the skeletal vibrations of C=C stretching mode. Furthermore, the absorption peaks at 1751–1764 cm^−1^ were representative of the C=O groups in carboxyl and carbonyl moieties. The absorption peaks at 1641 and 1639 cm^−1^ were ascribed to the δ(H_2_O) and C=N vibration groups (the G band refers to the first-order scattering of the E_2g_ phonon of *sp*^2^ C molecules), while the peak at 1415.75 cm^−1^ for ACWD corresponded to δ(CH_2_) and δ(CH_3_). The bands at 1004 and 1105 cm^−1^ were ascribed to υ(C=S). The new absorption peaks at 809 and 901.83 cm^−1^ were representative of υ(C–O–C). The disappearance and formation of new peaks of the vibration bands of the adsorbents are presented in Figure 2. 

#### 3.1.3. Scanning Electron Microscopy

The morphology of ACDW and LFB before and after ammonia uptake is shown in Figure 3. Figure 3A displays the SEM micrograph of ACDW before exposure to ammonia [31]; the cells formed agglomerates and had certain dimensions, with the existence of tiny macropores of different size at the external biomass surface. However, mineral substance could be shown within a conical shape with small pieces and irregular surface texture of the ACDW after adsorption of ammonia (Figure 3B). The morphology of ACDW after adsorption of ammonia was in the form of a helical tube of different sizes. Furthermore, the image showed the random distribution of highly porous regions, as presented in Figure 3B. Figure 3C displays the SEM image of LFB before adsorption, which showed a colonized surface with a homogeneous distribution, and the particles had certain dimensions [31]. However, mineral substance could be shown within a conical shape with small pieces and irregular surface texture of the LFB after adsorption of ammonia (Figure 3D). On the other hand, the mean diameter of ACDW before treatment of ammonia was 194.97 nm, but the average diameter ACDW after adsorption was 128.96 nm. The average diameter distribution of LFB before and after ammonia treatment was 176.09 and 144.16 nm, respectively.

### 3.2. Adsorption Studies

#### 3.2.1. Influence of pH

The pH plays a significant role in numerous cellular processes related to structure, energy metabolism, organelle function, enzymes, and proteins. The pH value affects the external charge of the adsorbents [48,49]. The adsorption of ammonia (NH_4_^+^) by two natural adsorbents of ACDW and LFB was examined at pH values of 2, 4, 6, 8, and 10 (Figure 4), while keeping the other parameters constant. The temperature of the solution, the volume of the solution, the amount of adsorbent used, the initial ammonia concentration, and the shaking period were 25 °C, 10 mL, 0.04 g, 30 mg·L^−1^, and 120 min, respectively. The obtained results indicated that the adsorption process of ammonia was favorable in acidic medium for LFB and alkaline medium for ACDW. It was found that the percentage of NH_4_ removal reached a maximum value at pH 6 with 63.3% and 67.8% for ACDW and FLB of dried microalgae, respectively, after which the percentage removal decreased slightly.

In the aquatic environment, ammonia volatile particles are considered to have a high solubility, because they simply exist in aquatic solutions. In aquaculture, the sum of the ionized form (NH_4_^+^) and unionized form (NH_3_) of nitrogen creates a buffer system of ammonium/ammonia. This equilibrium is based on the pH [28]. When the pH is less than 9.25, hydrogen ions are combined into ammonia to create ammonium ions, becoming the main species in the solution [10]. Therefore, as pH increases, the ammonia concentration rises. In natural waters, ammonium ions exist in greater concentrations than ammonia because of the prevalence of neutral pH. Hence, ammonia is characterized as the most poisonous N form, having a direct influence on the photosynthetic process of microalgal biomass [23]. Peterson et al. [50] showed that the active groups on the algae surface produce negative charges in acidic pH conditions, generating electrostatic interfaces across the cell surface and cationic species, which are responsible for adsorption.

#### 3.2.2. Influence of Contact Time

Contact time is inevitably a fundamental factor in all transfer phenomena such as adsorption, and the equilibrium time is one of the significant concerns for economical wastewater treatment [51]. The effect of contact time on the adsorption of NH_4_ by algae was evaluated in 0.04 g of adsorbent and 10 mL of 30 mg·L^−1^ NH_4_ solution, with a solution pH of 8 at 25 °C. The adsorption of NH_4_ by LFB from aqueous solution exhibited that the percentage removal increased to 63.95% with an increase in contact time. The uptake of NH_4_ by ACDW showed that most of the removal happened in the first 15 min of contact time (71.5%) (Figure 5). 

This augmentation could be attributed to the effect of the alteration in pH and stirring resulting from the addition of adsorbents [8]. The adsorption occurred in two steps. Initially, ions of ammonia were passively adsorbed onto the cell membranes of algae, where the sorption process was very fast because of an excessive number of vacant external sites. In the second step, sorption rates declined and eventually reached equilibrium due to the decrease in available sites, which were not easily occupied as a result of the repulsive powers among the solute particles adsorbed onto the bulk phase and the solid surface [36].

#### 3.2.3. Influence of the Adsorbent Dosage

The influence of ACDW and LFB was examined at 25 °C by varying the sorbent amounts from 0.02 to 0.1 g. For each of these runs, the concentration of ammonia was set to 30 mg·L^−1^, with 10 mL of 30 mg·L^−1^ NH_4_ solution having a pH of 8 for 120 min, and the results are demonstrated in Figure 6. The obtained results showed that the sorption of ammonia increased gradually with an increase in the quantity of ACDW or LFB from 0.02 g to 0.1 g, owing to the higher obtainability of binding sites of the surface area and a stronger driving force of ACDW and LFB [37].

#### 3.2.4. Influence of Initial NH_4_^+^ Concentration

In order to assess this influence, sorption experiments were performed at the initial NH_4_^+^ concentrations of 10, 20, 30, 40, 50, and 100 mg·L^−^^1^ at pH 8 with 0.04 g of adsorbent added to 10 mL solutions at 25 °C.

The present study showed that the maximum uptake of NH_4_^+^ was recorded at the concentration 10 mg·L^−1^, as demonstrated in Figure 7. At higher concentrations, the adsorption slightly decreased until the end of the tests at the same contact time and adsorption temperature. This phenomenon could be explained by the greatest number of existing adsorption sites of ACDW and LFB of dried *A. platensis*, which led to fast equilibrium and diffusion. Ion exchange takes place at the pores of the adsorbent material and its internal surface when the outside surface is saturated [52]. Furthermore, the ratio of NH_4_^+^ ions to active sites was small at low initial concentrations. In contrast, this ratio relatively increased at high initial NH_4_^+^ concentrations, where the number of NH_4_^+^ ions was greater than the number of active sites obtainable for sorption, which became exhausted owing to competition for actives sites. Therefore, a fast initial uptake of NH_4_^+^ ions occurred at the initial stage of adsorption process [53], showing a reduction in the removal efficiency at elevated ion concentrations. 

### 3.3. Applicability to Real Aquaculture Wastewater

Due to the large dimensions of *Arthrospira,* the separation of *Arthrospira* biomass (either ACDW or LFB) from the water column can be easily performed by filtering the water using a phytoplankton net (screen) with a mesh size of a 60 μm [54]. Many previous studies utilized microalgae as a bioremediation source for treating aquaculture wastewater such as *Chlorella vulgaris*, *Chlorococcum* sp., *Parachlorella kessleri*, *Scenedesmus quadricauda*, *Scenedesmus obliquus* [55], and *Arthrospira platensis* [24,56]. However, all previous studies cultivated living microalgae in the aquaculture wastewater. However, the present study is the first to use the complete dried biomass (ACDW) or lipid-free biomass (LFB) of *Arthrospira platensis* to remove ammonia from aquaculture wastewater. The results obtained showed that approximately 25.7% and 37.8% of ammonium ions were removed from aquaculture wastewater by ACDW and LFB, respectively. In contrast, the percentage of ammonium ion removal from synthetic aqueous solution achieved through dilution in distilled water with NH_4_Cl was 64.24% and 89.68% for ACDW and LFB, respectively (Table 1). The obtained results confirmed that the removal of ammonium ions by the two used adsorbents was affected by the real aquaculture wastewater. The effluents used in this work contain high concentrations of mixed effluents as pollutants resulting from pond effluents of Nile tilapia aquaculture wastewater, which resulted in competition with ammonia ions at the active sites, leading to lower removal efficiency. Different inorganic nitrogen types in wastewater can be converted to organic nitrogen by microalgae. Previous research has shown that nitrification or denitrification, as well as biological nitrogen uptake by distributed biomass, are the main mechanisms for nitrogen removal in algal systems [56,57,58]. Carbohydrates and polysaccharides in microalgal cell walls may contain several negatively charged (amino, hydroxyl, carboxyl, pyruvate, sulfide, phosphate, etc.) groups that act as active sites for adsorbing positively charged metal cations. Heavy metals could potentially be transferred into cells via the cell membrane, lowering their quantities in effluent [59]. Exopolysaccharides, commonly known as exopolysaccharides, are polysaccharides released by microalgae into the growth media [60]. Exopolysaccharide molecules with negatively charged groups can also absorb metals [61]. There are three main mechanisms (biodegradation, adsorption, consumption) that microalgae utilize to eliminate organics from wastewater. Microalgal cell walls feature numerous polymer groups that offer potential sorption sites for organic pollutants; however, the removal of organics by microalgal sorption was found to be rather low [62].

### 3.4. Isothermal Analysis

The obtained data in Figure 8, Figure 9 and Figure 10 agreed with the experimental results of Freundlich isotherm model, as confirmed by the high linear correlation coefficient (*R^2^* = 0.914 and 0.987 for ACDW and LFB, respectively), indicating that this model is favorable for physical adsorption. A strong bond occurred between NH_4_^+^ and the adsorbent, as shown by the value of 1/n, known as the heterogeneity factor. The deviation from sorption linearity is described as follows: if 1/n equals 1, the adsorption is linear, and the concentration of ammonia particles has no effect on the division between the two stages; when 1/n is less than 1, chemical adsorption occurs, resulting in a typical Langmuir isotherm (Table 2); when 1/n is greater than 1, cooperative adsorption occurs, resulting in an unusual Langmuir isotherm [51]. The values of factor “1/n” in this study were higher than one, indicating that a physical adsorption method on an external surface using this isotherm equation is preferable. When the Langmuir isotherm was applied, the obtained results agreed with the experimental data with high correlation coefficients (*R^2^* = 0.937 and 0.985 for ACDW and LFB, respectively). The equilibrium sorption (q_e_) increased with the rise in ammonia concentration. Many investigational models following a Langmuir equation include monolayer coverage. Furthermore, the Halsey isotherm model (Table 2) was not compatible with the adsorption technique, as the correlation coefficient (*R^2^*) was less than 0.900 for ACDW and LFB.

### 3.5. Kinetic Analysis

The kinetics of adsorbate uptake is critical for selecting the best operating conditions for design [49]. The kinetic data obtained from the adsorption of ammonium ions onto ACWD and LFB adsorbents was studied using the pseudo-first- and pseudo-second-order kinetic models. The kinetic parameters of ammonium ions are presented in Table 3. The linear correlation coefficients *R^2^* in the obtained data were not close to 1, implying that the pseudo-first-order (PFO) equation was not appropriate for exposing the reaction mechanism of ammonium ions with the adsorbents. In contrast, it can be seen that the pseudo-second-order (PSO) kinetic equations fitted considerably well with the investigational data, as shown in Table 3. The correlation coefficient values were close to one, indicating good agreement (*R^2^* = 0.999 and 0.996 for ACDW and LFB, respectively). Furthermore, the anticipated (q_e_ calculation) values were close to the experimental ones (q_e_ experimental). Thus, the PSO equation corresponded well to the ammonium ion adsorption method exhibited by ACDW and LFB (Figure 11 and Table 3). 

### 3.6. Bacterial Count

Today, algal cells are recognized to produce many water-soluble natural pigments such as phycobiliproteins, carotenoids, phycoerythrin, and chlorophyll, which are important in many industries as antimicrobial and antioxidant agents [63], biofertilizer [64,65,66,67], aquaculture [68,69,70], food supplements [71], pharmaceuticals and nutraceuticals [72], and bioremediation [31,73]. There are many types of algae that are broadly known to produce intra- and extracellular metabolites with activity against fungal, bacterial, and viral pathogens [74], due to their high content of lipids and fatty acids [5,75,76]. The present work evaluated the effects of different forms of *A. platensis* (ACDW and LFB) on the bacterial count of aquaculture wastewater (RAW) on agar media. The results of treating aquaculture wastewater (RAW) with LFB and ACDW in terms of the bacterial count of each group are presented in Table 4 and Figure 12. The number of bacterial cells was high in all tested groups, exceeding 300 CFU, except for the LFB-RAW group, with <100 CFU (Figure 12 and Table 4).

Compared to LFB-RAW (which had a lower bacterial count of <100 CFU), all other tested groups showed a bacterial count of >300 CFU. This finding may be attributed to the lipid extraction process from *A. platensis* biomass, whereas LFB-RAW was treated by LFB which was a biodiesel byproduct of *A. platensis* biomass. This result indicates that the extraction of lipids from *Arthrospira* improved its efficiency and ability to inhibit bacterial growth and multiplication on agar media. The extraction process is well known to increase the surface area of the algal cell due to disruption of the cell walls, resulting in the internal organelles and structures such as the cytoplasm being available on the external surfaces. Moreover, this mechanism leads to greater involvement and attachment of water to the intracellular algal voids, intercellular constituents, and superficial algal surfaces. This increases the exposure area of biomass, resulting in improved bacterial inhibition. Previously, *A. platensis* was evaluated for many medical and pharmaceutical applications due to its intracellular and extracellular polysaccharide molecules which have potential activities against various viral infections including HIV and human cytomegalovirus [77]. Moreover, *A. platensis* extracts showed a high inhibitory effect against bacterial species such as different *Vibrio* species (*V. anguillarum*, *V. lentus*, *V. parahaemolyticus*, *V. alginolyticus*, *V. splendidus*, and *V. scophthalmi*), different *Streptococcus* species (*S. pyogenes* and *S. aureus*), *E. coli*, and *P. aeruginosa* [78], in addition to antifungal properties against *C. albicans* and *Aspergillus flavus* [79].

### 3.7. Rotifer Experiment

Rotifer *B. plicatilis* is the biggest zooplankton group in water ecosystems and is of particular importance as a live feed in marine hatcheries [80,81]. In aquatic ecosystems, the zooplankton population can be influenced by many environmental conditions such as temperature, oxygen demand, pH, predation, and competition [82,83], as well as nutrient loading [84]. Figure 13 shows the rotifer population growth, rotifer mortality, rotifer female egg-carrying population, and the mortality of rotifer female egg-carrying population using different concentrations of the studied biomass. 

Compared with the control levels of *A. platensis* (LFB-Control and ACDW-Control), all levels of *A. platensis* saturated with ammonium ions that were removed from aquaculture wastewater led to a reduction in the rotifer population, as well as a decrease in the population of females carrying eggs (Figure 13a,b), whereas it increased the total rotifer mortality and the mortality of females carrying eggs (Figure 13c,d). At all LFB and ADCW levels, the highest rotifer population and population of females carrying eggs was achieved at 0.05 g LFB (26,000 and 1900 ind.·L^−1^, respectively) and 0.10 g ACDB (25,700 and 4435 ind.·L^−1^, respectively). On the other hand, the lowest rotifer population and population of females carrying eggs were achieved at 0.20 g of either LFB (15,500 and 1150 ind.·L^−1^, respectively) or ACDB (14,300 and 3500 ind.·L^−1^). 

To the best of our knowledge, the current study is the first work reporting the potential of ammonia-loaded microalgae biomass as a feed source for rotifers. The study concluded that the sensitivity of marine rotifer *B. plicatilis* to ammonia concentration was due to a reduced rotifer population and population of females carrying eggs, as well as increased rotifer mortality. Due to ammonia from nutrients in the marine environment influencing marine life, zooplankton communities respond to a wide variety of disturbances, including nutrient loading [84].

### 3.8. Comparison of NH_4_-N Elimination by Various Species of Microalgae

According to recent studies, microalgal systems can efficiently treat many kinds of ammonium-containing wastewaters, as presented in Table 5.

## 4. Conclusions

This study examined the potential application of *Arthrospira platensis* NIOF17/003 complete dry weight (ACDW) and lipid-free biomass (LFB) as low-cost, readily available, highly stable, and environmentally friendly sorbents for the removal of ammonium ions (NH_4_^+^) from synthetic aqueous solution, as well as its applicability to real aquaculture wastewater. In addition, this study evaluated the effect of the adsorbents ACDW and LFB on the bacterial count in aquaculture wastewater. Furthermore, the use of ammonia-laden ACDW and LFB was examined as a feed source for a marine rotifer (*Brachionus plicatilis*). The adsorption of ammonia onto *A. platensis* in a synthetic aqueous solution was examined using a batch system with respect to the initial ammonia concentration, adsorbent dose, pH, and contact time in the aqueous solution. Maximum removal was found after 2 h at pH 8 and 6 for ACWD and LFB, respectively, with an optimal adsorbent dose of 0.1 g at 10 mg·L^−1^ ammonium ions. However, the removal of ammonium using ACDW and LFB in the synthetic aqueous solution (64.24% and 89.68%, respectively) was higher than that of the real aquaculture effluents (25.70% and 37.80%, respectively). The maximum ammonium ion adsorption capacity (q_max_ = 0.379 and 0.745 mg·g^−1^) was obtained for ACWD and LFB, respectively. The Freundlich and Langmuir models agreed well with the equilibrium sorption results. The obtained data exhibited that the appropriate kinetic models for both ACWD and LFB were well described with high correlation coefficients by the second-order equations. On the other hand, as a result of the treatment of aquaculture wastewater (RAW) with LFB and ACDW, the bacterial count of LFB, ACDW, ACDW-RAW, and RAW groups was >300 CFU, whereas that for LFB-RAW as <100 CFU. The study concluded that *B. plicatilis* was sensitive to *A. platensis* biomass loaded with ammonia. Lastly, the results in this work confirmed that the biomass of *A. platensis* is a promising candidate for ammonia removal from aquaculture wastewater, with potential utilization as a dry feed for rotifer *B. plicatilis* in marine hatcheries.

## Figures and Tables

**Figure 1 materials-14-05460-f001:**
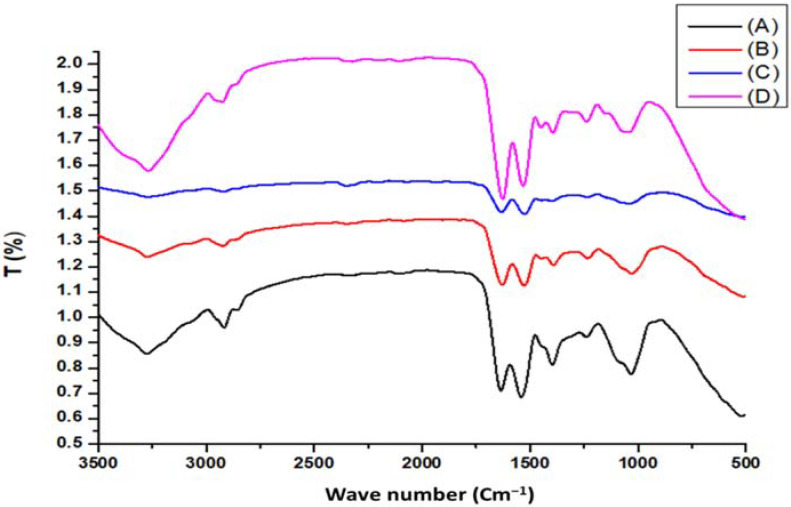
FTIR spectra for (A) ACDW before treatment, (B) ACDW after treatment, (C) LFB before adsorption, and (D) LFB after adsorption of ammonia.

**Figure 2 materials-14-05460-f002:**
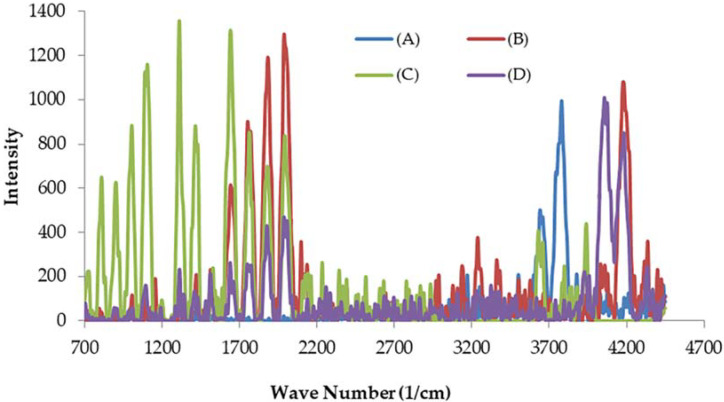
Raman spectrum for spectra for (A) ACDW before treatment, (B) ACDW after treatment, (C) LFB before treatment, and (D) LFB after treatment of ammonia.

**Figure 3 materials-14-05460-f003:**
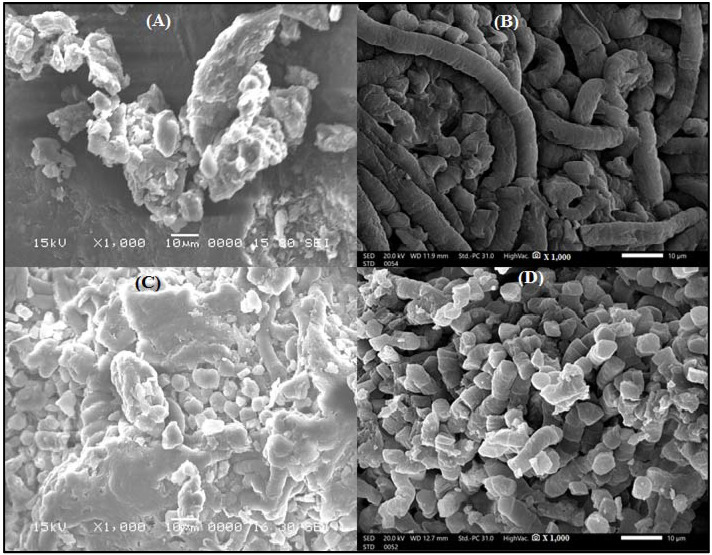
SEM pictures of ACDW before adsorption (**A**), ACDW after adsorption (**B**) and LFB before adsorption (**C**), and LFB after adsorption (**D**).

**Figure 4 materials-14-05460-f004:**
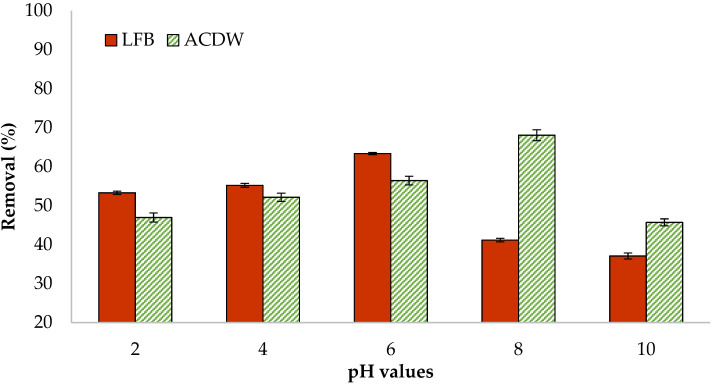
Influence of pH value on the adsorption of ammonia using whole biomass (ACDW) and lipid-free biomass (LFB) of *Arthrospira platensis*.

**Figure 5 materials-14-05460-f005:**
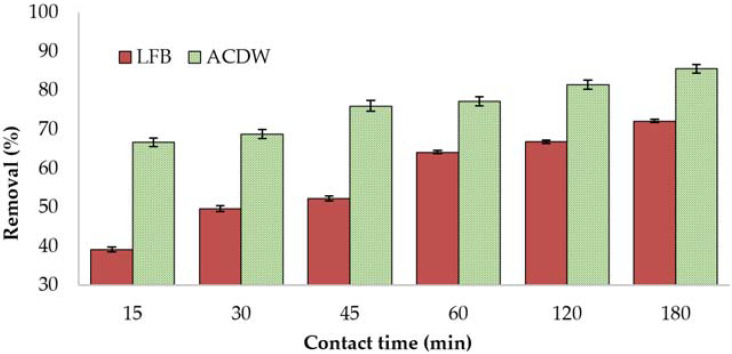
Influence of contact time on the treatment of ammonia using whole biomass (ACDW) and lipid-free biomass (LFB) of *Arthrospira platensis*.

**Figure 6 materials-14-05460-f006:**
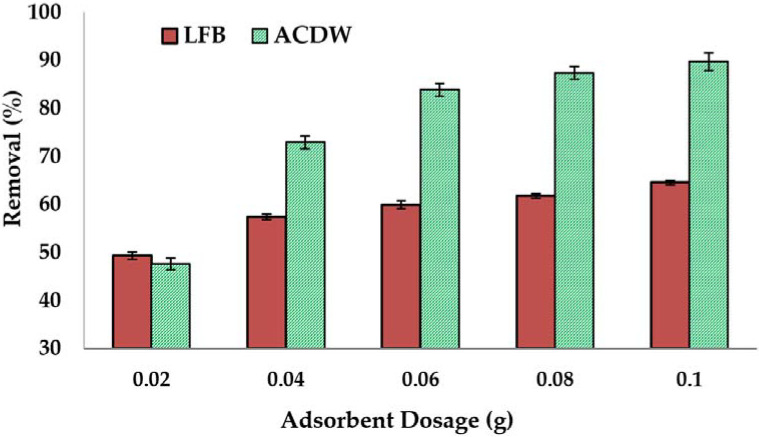
Influence of adsorbent dosage on the treatment of ammonia using whole biomass (ACDW) and lipid-free biomass (LFB) of *Arthrospira platensis*.

**Figure 7 materials-14-05460-f007:**
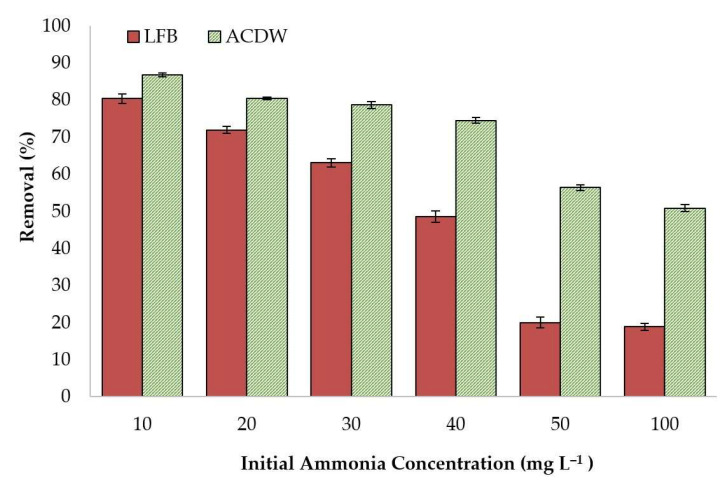
Influence of initial ammonia concentration on the treatment of ammonia using whole biomass (ACDW) and lipid-free biomass (LFB) of *Arthrospira platensis*.

**Figure 8 materials-14-05460-f008:**
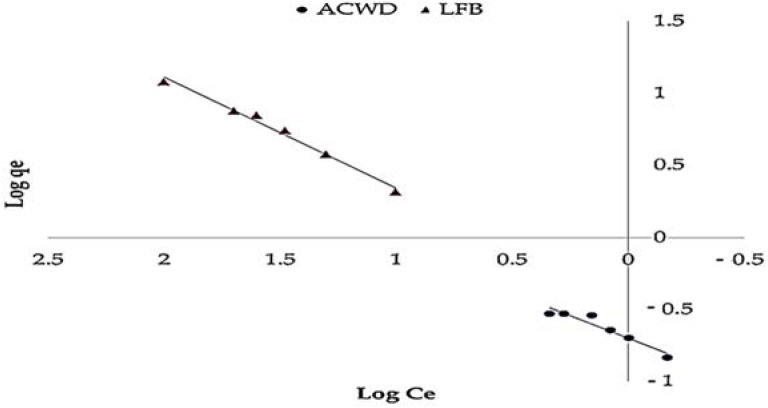
Freundlich isotherm plot for the sorption of ammonia using whole biomass (ACDW) and lipid-free biomass (LFB) of *Arthrospira platensis*.

**Figure 9 materials-14-05460-f009:**
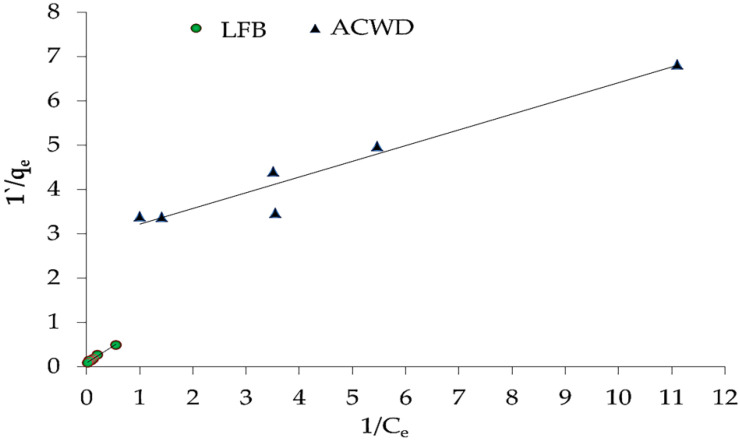
Langmuir isotherm plot for the sorption of ammonia using whole biomass (ACDW) and lipid-free biomass (LFB) of *Arthrospira platensis*.

**Figure 10 materials-14-05460-f010:**
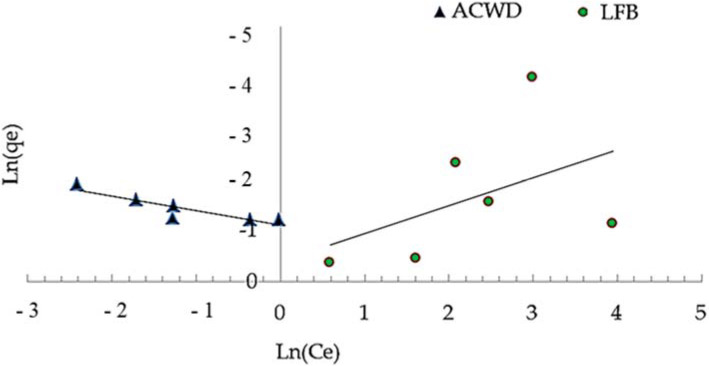
Halsey model plot for the sorption of ammonia using whole biomass (ACDW) and lipid-free biomass (LFB) of *Arthrospira platensis*.

**Figure 11 materials-14-05460-f011:**
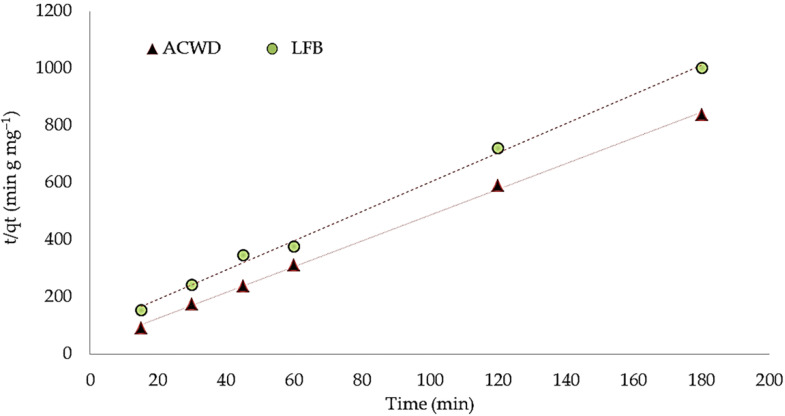
Pseudo-second-order kinetics of ammonium using whole biomass (ACDW) and lipid-free biomass (LFB) of *Arthrospira platensis*.

**Figure 12 materials-14-05460-f012:**
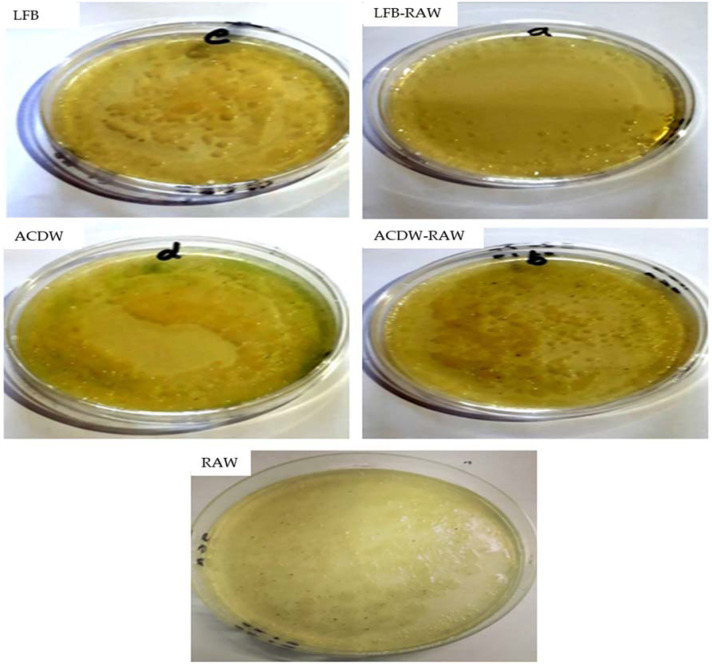
Bacterial count on agar media of all groups: LFB: lipid-free biomass, LFB-RAW: real aquaculture wastewater treated with LFB, ACDW: *Arthrospira* complete dry weight, ACDW-RAW: real aquaculture wastewater treated with ACDW, RAW: real aquaculture wastewater.

**Figure 13 materials-14-05460-f013:**
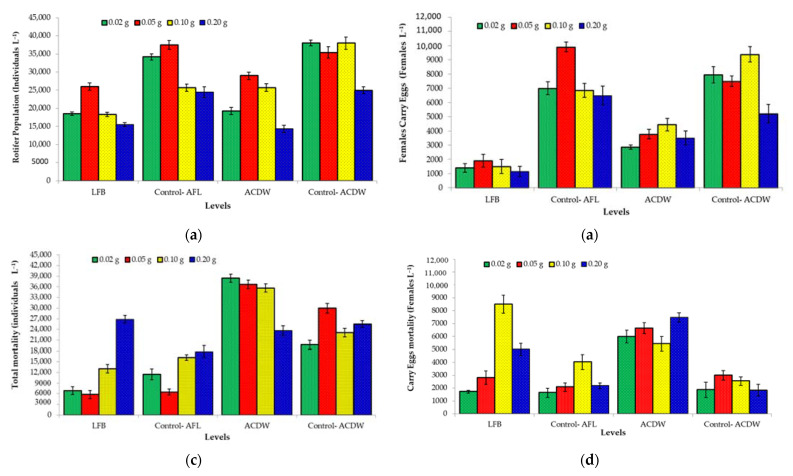
Effect of several levels of cyanobacterial strain *Arthrospira platensis* NIOF17/003 (LFB and ACDW) saturated with ammonium ions removed from aquaculture wastewater, compared to the same traditional levels of *A. platensis* as control (LFB-Control and ACDW-Control) on the *Brachionus plicatilis* population, the population of females carrying eggs, total rotifer mortality, and mortality of the females carrying eggs ((**a**–**d**), respectively).

**Table 1 materials-14-05460-t001:** Treatment for ammonium ions using ASDW and LFB in different water samples.

Types of Water	Ammonium Ions Removal (%)	
	ACDW	LFB
Synthetic aqueous solution	64.24	89.68
Real aquaculture effluent	25.70	37.80

**Table 2 materials-14-05460-t002:** Isotherm factor of ammonium ion adsorptions (ACDW and LFB).

Isotherm Models	Parameters	ACDW	LFB
Freundlich Model	*R^2^*	0.914	0.987
	1/n	1.463	1.279
	K_f_	10.73	3.16
Langmuir Model	*R^2^*	0.937	0.985
	q_max_ (mg·g^−1^)	0.379	0.745
Halsey Model	*R^2^*	0.884	0.213
	1/n_H_	2.797	0.390
	K_H_	2.884	1.64

**Table 3 materials-14-05460-t003:** The kinetic factors for the removal of ammonium ions by ACDW and LFB.

First-Order Kinetic			Second-Order Kinetic		
Adsorbent	K_1_ (1 min)	*R^2^*	q_e_ Calc. (mg·g^−1^)	K_2_ (g·mg^−1^·min^−1^)	*R^2^*
ACWD	122.24	0.337	1.65	0.604	0.999
LFB	0.0034	0.239	3.379	0.295	0.996

**Table 4 materials-14-05460-t004:** Bacterial count as a result of treatment of real aquaculture wastewater using LFB and ACDW.

Bacterial Count	Groups
Higher than 300 CFU	LFB
Less than 100 CFU	LFB-RAW
Higher than 300 CFU	ACDW
Higher than 300 CFU	ACDW-RAW
Higher than 300 CFU	RAW

LFB: lipid-free biomass, LFB-RAW: real aquaculture wastewater treated with LFB, ACDW: *Arthrospira* complete dry weight, ACDW-RAW: real aquaculture wastewater treated with ACDW, RAW: real aquaculture wastewater.

**Table 5 materials-14-05460-t005:** Comparison of NH_4_-N removal by different species of microalgae.

Species (Microalgae)	NH_4_-N Removal	Reference
*Spirulina (Arthrospira)*	84–96%	[85]
*Spirulina* sp.	79%	[86]
*Spirulina platensis*	80%	[87]
*Chlorella* sp.	18%	[11]
*Scenedesmus* sp.	71–92.8%	[88]
*Scenedesmus obliquus*	66–73%	[27]
*Chlorella sorokiniana*	75%	[89]
*Arthrospira platensis* (ACDW)	64%	This study
*Arthrospira platensis* (LFB)	89%	This study

## Data Availability

The data presented in this study are available on request from the corresponding authors.

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
