# Peer review of "Ammonia Bioremediation from Aquaculture Wastewater Effluents Using *Arthrospira platensis* NIOF17/003: Impact of Biodiesel Residue and Potential of Ammonia-Loaded Biomass as Rotifer Feed"

_materials, 2021, doi:10.3390/ma14185460_

Round 1

Reviewer 1 Report

Manuscript ID: materials-1370139

 The paper "Ammonia Bioremediation from Aquaculture Wastewater Effluents using Arthrospira platensis NIOF17/003: Impact of Biodiesel Residue and Potential of Ammonia-Loaded Biomass as Rotifer Feed" presents the results of a study on the efficiency of ammonium ion removal from synthetic NH4Cl solution and from aquaculture wastewater using Arthrospira platensis and lipid-free biomass.

The efficiency of ammonium removal was studied under the following parameters: contact time, initial ammonium ion concentration, pH and adsorbent dose.

The study also evaluated the effect of the adsorbents on the bacterial counts in the aquaculture wastewater and the use of ammonia-laden adsorbents was examined as a feed source for marine rotifer.

The isotherm and kinetic model were used to describe adsorption process.

The equilibrium adsorption isotherms were estimated by Freundlich, Langmuir and Halsey models. The experimental kinetic study was suitably fitted by a pseudo-second order equation.

The article is quite interesting, well written and the results of the research are clearly presented. In general, in my opinion, the manuscript meets the standards of the journal Materials. Nevertheless, some elements of the paper should be improved.

Specific Comments:

- Line 21: 140: should be further clarified, ion exchange or adsorption?

- Section "Application of Aquaculture Wastewater" - More physicochemical parameters of the aquaculture effluent sample should be supplemented, for example: alkalinity, TOC or COD.

- Line 144: "The mixture was shaken at 120 rpm..." It should be clarified - mixing or shaking?

- Line 363: It should be clarified - NH3 or NH4 was tested?

- Section "Influence of contact time". The pH of the testing sample should be given.

- Figure 6 "Adsorbent Dosage" - Please correct - oo.8 > 00.8

- Information on the reagents used should be supplemented.

- The method of determination of the ammonium ion should be described.

- The description of the test results should be improved by comparison with the results of similar studies.

- It would also be valuable to indicate how used sorbents can be utilised.

Author Response

SUMMARY OF AUTHOR(S) RESPONSE TO REVIEWER’S COMMENTS

Manuscript Title: Ammonia Bioremediation from Aquaculture Wastewater Effluents using Arthrospira platensis NIOF17/003: Impact of Bio-diesel Residue and Potential of Ammonia-Loaded Biomass as Rotifer Feed

Authors: Mohamed Ashour; Ahmed E. Alprol; Ahmed M.M. Heneash; Hosam Saleh; Khamael M. Abualnaja; Dalal Alhashmialameer; Abdallah Tageldein Mansour

Reviewer 2# Round 1 Comment

Author(s) response

Comments and Suggestions for Authors

The paper "Ammonia Bioremediation from Aquaculture Wastewater Effluents using Arthrospira platensis NIOF17/003: Impact of Biodiesel Residue and Potential of Ammonia-Loaded Biomass as Rotifer Feed" presents the results of a study on the efficiency of ammonium ion removal from synthetic NH4Cl solution and from aquaculture wastewater using Arthrospira platensis and lipid-free biomass.The efficiency of ammonium removal was studied under the following parameters: contact time, initial ammonium ion concentration, pH and adsorbent dose. The study also evaluated the effect of the adsorbents on the bacterial counts in the aquaculture wastewater and the use of ammonia-laden adsorbents was examined as a feed source for marine rotifer.The isotherm and kinetic model were used to describe adsorption process. The equilibrium adsorption isotherms were estimated by Freundlich, Langmuir and Halsey models. The experimental kinetic study was suitably fitted by a pseudo-second order equation.

The article is quite interesting, well written and the results of the research are clearly presented. In general, in my opinion, the manuscript meets the standards of the journal Materials. Nevertheless, some elements of the paper should be improved.

The authors would like to thank Reviewer # 2 for his kind and his interesting and valuable comments. All Reviewer # 2 comments have been considered carefully by the authors. These comments significantly improve the manuscript. 

Line 21: 140: should be further clarified, ion exchange or adsorption?

Paragraph about adsorption was clarified (Page: 3; Lines: 104-109).

Section "Application of Aquaculture Wastewater" - More physicochemical parameters of the aquaculture effluent sample should be supplemented, for example: alkalinity, TOC or COD.

The water sample was consumed.

Line 144: "The mixture was shaken at 120 rpm..." It should be clarified - mixing or shaking?

Corrected (Page: 3; Line: 149).

Line 363: It should be clarified - NH3 or NH4 was tested?

Corrected (Page: 11; Line: 379).

Section "Influence of contact time". The pH of the testing sample should be given.

Section "Influence of contact time". The pH of the testing sample was added (Page: 11; Lines: 374-377).

Figure 6 "Adsorbent Dosage" - Please correct - oo.8 > 00.8.

oo.8 in  Figure 6 "Adsorbent Dosage was corrected (Page: 13; Line: 426).

Information on the reagents used should be supplemented.

Reagents used were supplemented (Page: 6; Lines: 286-289).

The method of determination of the ammonium ion should be described.

The method of determination of the ammonium ion was added (Page: 4; Lines: 153-158).

The description of the test results should be improved by comparison with the results of similar studies.

Improved (Page: 20 and 21; Lines 220-224).

It would also be valuable to indicate how used sorbents can be utilized.

Added (Page: 14 and 15; Lines 468-482).

We would like to extend our sincere thanks and appreciation to the reviewers and editorial board. In fact, their comments and guidance added a lot to the research and increased its scientific content. Therefore, the words cannot express their gratitude for their time and effort they put in evaluating this research.

Reviewer 2 Report

This is a review paper, "Ammonia Bioremediation from Aquaculture Wastewater Effluents using Arthrospira platensis NIOF17/003: Impact of Biodiesel Residue and Potential of Ammonia-Loaded Biomass as 4 Rotifer Feed", studied the capability of A. Platensis for removal of ammonium ions from aquaculture wastewater discharge. The work is well written, and discussions in each section provide a clear explanation about the frame of the work. As such it is a valuable contribution and the paper should be published after minor revision.

I have, some comments to the manuscript as follow:

  • Keep the same format. Use "Figure" instead of "Fig." in the whole manuscript.

  • Page 2, Line 81:

11–13 ML–1 Ha–1 year–1 …. Please correct the unit.

  • Page 4, Line 74: Please add the equation number for the second equation of the Langmuir model:           

  • Page 6, Line 251:

Please correct the equation number.

  • Page 7, Line 318:

S. platensis ….. It should be: A. platensis.

  • Pages 8, Line 327:

… while he peak at…. It should be “the”.

  • Page 8, Lines 329, 330:

Please check these numbers 553.39 cm1 and 78.77 cm1 and 74.12 cm1. The Raman spectras in Figure 2 are started from 700 cm1. Check the numbers, and if they are correct, then you need to replot the figure.

  • Page 10, 3.2.1. Influence of pH:

What are the other fixed parameters during the evaluation of pH? for example what is the contact time, adsorbent dosage,…

  • Page 10, 3.2.2. Influence of contact time:

Similar to the previous section, please add the other fixed reaction parameters during this evaluation.

  • Page 11, 3.2.3. Influence of the Adsorbent Dosage:

Similar to the previous section, please add the other fixed reaction parameters during this evaluation.

  • Page 12, 3.2.4. Influence of initial NH4+ concentration:

Similar to the previous section, please add the other fixed reaction parameters during this evaluation.

  • Page 15, Lines 493, 494:

Please in the first place, define what PFO and PSO are standing for.

pseudo-first-order (PFO)

pseudo-second-order (PSO)

Author Response

SUMMARY OF AUTHOR(S) RESPONSE TO REVIEWER’S COMMENTS

Manuscript Title: Ammonia Bioremediation from Aquaculture Wastewater Effluents using Arthrospira platensis NIOF17/003: Impact of Bio-diesel Residue and Potential of Ammonia-Loaded Biomass as Rotifer Feed

Authors: Mohamed Ashour; Ahmed E. Alprol; Ahmed M.M. Heneash; Hosam Saleh; Khamael M. Abualnaja; Dalal Alhashmialameer; Abdallah Tageldein Mansour

Reviewer 2# Round 1 Comment

Author(s) response

Comments and Suggestions for Authors

This is a review paper, "Ammonia Bioremediation from Aquaculture Wastewater Effluents using Arthrospira platensis NIOF17/003: Impact of Biodiesel Residue and Potential of Ammonia-Loaded Biomass as Rotifer Feed", studied the capability of A. Platensis for removal of ammonium ions from aquaculture wastewater discharge. The work is well written, and discussions in each section provide a clear explanation about the frame of the work. As such it is a valuable contribution and the paper should be published after minor revision.

I have, some comments to the manuscript as follow:

The authors would like to thank Reviewer # 2 for his kind and his interesting and valuable comments. All Reviewer # 2 comments have been considered carefully by the authors. These comments significantly improve the manuscript. 

Keep the same format. Use "Figure" instead of "Fig." in the whole manuscript.

Replaced in the whole manuscript.

Page 2, Line 81:

11–13 ML–1 Ha–1 year–1   Please correct the unit.

Corrected (Page: 2;  line: 81).

 Page 4, Line 74: Please add the equation number for the second equation of the Langmuir model:          

Page 6, Line 251: Please correct the equation number.

The equations numbers were added and corrected (Pages:  4 and 5).

 Page 7, Line 318: S. platensis … It should be: A. platensis.

Corrected (Page: 8; line: 333).

Pages 8, Line 327: while he peak at…. It should be “the”.

Corrected (Page: 8; line: 342).

·         Page 8, Lines 329, 330:

Please check these numbers 553.39 cm1 and 78.77 cm1 and 74.12 cm1. The Raman spectras in Figure 2 are started from 700 cm1. Check the numbers, and if they are correct, then you need to replot the figure.

These numbers were deleted from section of Raman spectra, due to the drawing does not appear well in range less from 500 cm–1 as attached figure.

·        Page 10, 3.2.1. Influence of pH: What are the other fixed parameters during the evaluation of pH? for example what is the contact time, adsorbent dosage,…

The other fixed parameters during the evaluation of pH were added (Page: 11; lines: 374-377).

·         Page 10, 3.2.2. Influence of contact time:

Similar to the previous section, please add the other fixed reaction parameters during this evaluation.

The other fixed parameters during the estimation of contact time were added in manuscript (Page: 11 and 12; lines: 401-403).

·        Page 11, 3.2.3. Influence of the Adsorbent Dosage:

Similar to the previous section, please add the other fixed reaction parameters during this evaluation.

The other fixed parameters during the assessment of adsorbent dosage were added in manuscript (Page: 12; lines: 419-422).

·         Page 12, 3.2.4. Influence of initial NH4+ concentration:

Similar to the previous section, please add the other fixed reaction parameters during this evaluation.

The other fixed parameters during the assessment of initial NH4+ concentration were added in manuscript (Page 13 lines 430-432).

·        Page 15, Lines 493, 494: Please in the first place, define what PFO and PSO are standing for pseudo-first-order (PFO) pseudo-second-order (PSO).

PFO and PSO were defined (Page: 17; lines: 528 and 530).

We would like to extend our sincere thanks and appreciation to the reviewer (s) and editorial board. In fact, their comments and guidance added a lot to the research and increased its scientific content. Therefore, the words cannot express their gratitude for their time and effort they put in evaluating this research.
